# Sensing and Microbiological Activity of a New Blue Fluorescence Polyamidoamine Dendrimer Modified with 1,8-Naphthalimide Units

**DOI:** 10.3390/molecules29091960

**Published:** 2024-04-25

**Authors:** Ivo Grabchev, Albena Jordanova, Evgenia Vasileva-Tonkova, Ivan L. Minkov

**Affiliations:** 1Faculty of Medicine, Sofia University “St. Kliment Ohridski”, 1407 Sofia, Bulgaria; benita1968@abv.bg (A.J.); minkov.ivan@gmail.com (I.L.M.); 2The Stephan Angeloff Institute of Microbiology, Bulgarian Academy of Sciences, 1113 Sofia, Bulgaria; evaston@yahoo.com

**Keywords:** 1,8-naphthalimide, photophysics, sensors, dendrimer, antimicrobial

## Abstract

A novel second-generation blue fluorescent polyamidoamine dendrimer peripherally modified with sixteen 4-*N,N*-dimethylaninoethyloxy-1,8-naphthalimide units was synthesized. Its basic photophysical characteristics were investigated in organic solvents of different polarity. It was found that in these solvents, the dendrimer is colorless and emitted blue fluorescence with different intensities depending on their polarity. The effect of the pH of the medium on the fluorescence intensity was investigated and it was found that in the acidic medium, the fluorescence is intense and is quenched in the alkaline medium. The ability of the dendrimer to detect metal ions (Pb^2+^, Zn^2+^, Mg^2+^, Sn^2+^, Ba^2+^, Ni^2+^, Sn^2+^, Mn^2+^, Co^2+^, Fe^3+^, and Al^3+^) was also investigated, and it was found that in the presence of Fe^3+^, the fluorescent intensity was amplified more than 66 times. The antimicrobial activity of the new compound has been tested in vitro against Gram-positive *B. cereus* and Gram-negative *P. aeruginosa*. The tests were performed in the dark and after irradiation with visible light. The antimicrobial activity of the compound enhanced after light irradiation and *B. cereus* was found slightly more sensitive than *P. aeruginosa*. The increase in antimicrobial activity after light irradiation is due to the generation of singlet oxygen particles, which attack bacterial cell membranes.

## 1. Introduction

In recent years, antimicrobial resistance has been among the leading causes of death worldwide and a serious threat to public health. It is caused by the excessive and uncontrolled use of antibiotics and antimicrobials used in medical practice, which leads to resistance to their treatment by pathogenic bacteria, fungi, viruses, and parasites as a result of extremely rapid mutations and adaptation processes [1,2,3].

Antimicrobial photodynamic therapy (APDT) is a promising approach to treating bacterial infections that are not susceptible to antibiotics [4,5,6,7]. This method is based on using light to activate exogenous compounds called photosensitizers (PSs), generating reactive oxygen species that kill pathogenic microorganisms [8,9]. Photosensitizers have very little antimicrobial activity in the dark. Under the action of light of an appropriate wavelength, in which the photosensitizing compound absorbs photons, typically in the visible spectral range (400–700 nm), it transitions from a ground state through an excited singlet state to a triplet state in which it reacts with triplet oxygen molecules. The photoactivation of the PS results in the formation of reactive oxygen species (ROS) including singlet oxygen (^1^O_2_) or radicals that are toxic to target cells. A PS located in bacteria or on the bacterial surface induces cell death [10,11,12].

With the development of synthetic organic chemistry, a new type of three-dimensional macromolecules called dendrimers, having monodispersity and perfectly branched symmetrical structure, are being developed and studied [13,14,15,16,17]. They combine the properties of low- and high-molecular substances, which is why the interest in this class of compounds is constantly growing. Fluorescent dendrimers containing chromophore fragments in specific, predetermined positions of the molecule have been developed, which leads to the emergence of new properties and new areas of application in chemistry, biology, pharmacy, genetic engineering, nanomedicine, etc. [18,19,20,21,22,23,24]. Dendrimer molecules contain a large number of closely spaced functional groups in their structure. If these groups are functionalized with biologically active substances, an increase in the biological activity of dendrimers can be expected compared to low molecular weight compounds.

New compounds with different chemical structures are currently being intensively investigated to discover highly effective antimicrobial preparations [25]. In this respect, cyclic imides are of particular interest due to their very well-pronounced biological activity. 1,8-Naphthalimide derivatives are a type of cyclic imides characterized by hydrophobicity and a large π-compressed aromatic skeleton, which can readily interact with biological systems via noncovalent interactions with well-pronounced antimicrobial and antitumor activity [26,27,28,29,30]. 

Substituted at the C-4 position 1,8-naphthalimides can bind peripherally in the dendrimer structures. Thus, dendrimer molecules are given new valuable properties and expand the type of fields of application such as sensors, biologically active substances with antitumor and microbiological activity, imaging diagnostics, etc. [31,32,33,34].

This study aims to synthesize a new polyamidoamine (PAMAM) dendrimer modified with sixteen 4-*N,N*-dimethylaninoethyloxy-1,8-naphthalimide units and to investigate its photophysical characteristics. The ability of the dendrimer to detect metal ions and protons by a change in fluorescence intensity was evaluated. Its microbiological activity against Gram-positive *Bacillus cereus* and Gram-negative *Pseudomonas aeruginosa* used as model bacterial strains in the dark and after light irradiation has also been investigated.

## 2. Results and Discussion

### 2.1. Synthesis of Dendrimer

The modified with 4-nitro-1,8-naphthalimide units PAMAM dendrimer has been used as the initial material for the synthesis of blue-emitting fluorescent dendrimer (Figure 1). The nitro group has well-expressed electron acceptor properties. Its presence in the C-4 position of the 1,8-naphthalimide structure does not allow donor–acceptor interaction and polarization of the molecule and such dendrimers do not exhibit fluorescent properties. Therefore, it was necessary to replace it with electron-donating groups such as amino or alkoxy groups. The structure fluorophore-spaser-receptor was used in the design of the dendrimer. An *N,N*-dimethylaminoethyloxy group (-OCH_2_CH_2_N(CH_3_)_2_) attached to the C-4 atom of the 1,8-naphthalimide core was used as a receptor fragment. This fragment contains a tertiary nitrogen atom with a lone pair of electrons and allows photoinduced electron transfer to occur. On the other hand, this nitrogen atom is capable of protonation and formation of complexes with various metal cations [35,36]. Nucleophilic substitution of the nitro group with *N*,*N*-dimethylaminoethyloxy was carried out in DMF solution at 50 °C for 8 h, after which the homogeneous solution was poured into ice water and the precipitate formed was filtered. The resulting light-yellow product was filtered, dried, and used without further purification to study the photophysical characteristics and microbiological activity.

### 2.2. Infrared Characterization of Dendrimer

Figure 1 compares the IR spectra of the initial dendrimer containing nitro (-NO_2_) groups and dendrimer after its nucleophilic substitution with alkoxy (-OCH_2_CH_2_N(CH_3_)_2_) groups. The carboxyl groups C=O of the 1,8-naphthalimide structure have two characteristic absorption bands, the positions of which strongly depend on the nature of the substituent in the 1,8-naphthalimide structure and its polarization. These vibrational bands are due to the asymmetric and symmetric vibrations of the carbonyl groups, which are induced by the imide resonance structures. The two typical absorption bands for the asymmetric and symmetric vibration of the carbonyl groups of the 4-nitro-substituted 1,8-naphthalimide are seen, which are at 1706 cm^−1^ and 1662 cm^−1^and their characteristic shift after the replacement of the nitro group by the alkoxy group, respectively at 1684 cm^−1^ and 1651 cm^−1^. The intense band at 1524 cm^−1^ is due to the asymmetric vibration induced by the nitro group. After its replacement with (-OCH_2_CH_2_N(CH_3_)_2_), characteristic vibrations of the nitro group disappear, and the position of C=O groups is shifted.

Also, in the IR spectrum of the dendrimer, new characteristic absorption bands appear with weak intensity at 1268 cm^−1^ and 1082 cm^−1^ due to the ether groups Ar-O-CH_2_- from the 1,8-naphthalimide structure. The peaks at 1340–1354 cm^−1^ and 1282–1284 cm^−1^ were attributed to the imide group (C-N-C) of the 1,8-naphthalimide structures and the C-N bonds from the tertiary amino groups (-N(CH_3_)_2_). The characteristic out-of-plane deformation vibrations of the aromatic structure of the naphthalene structure are in the region of 758–785 cm^−1^.

### 2.3. Photophysical Characterization

The spectral characteristics of the modified PAMAM dendrimer with 4-*N,N*-dimethylaminoethoxy-1,8-naphthalimides in a solution of eight organic solvents of different polarities were investigated and the results were collected in Table 1.

The long-wavelength absorption band in the UV region is a charge transfer band due to π→ π* electron transfer at the S_0_→S_1_ transition. In the investigated solvents, the dendrimer is colorless and absorbed in the near UV region with maxima at 342–360 nm. A well-pronounced solvatochromism with a hypsochromic shift (Δλ_A_ = 18 nm) was observed in the transition from polar to nonpolar solvents. The emitted fluorescence is blue with maxima in the region 421–440 nm, the position of which also depends on the polarity of the solvents (Figure 2). Probably, in alcohol solutions, due to the possibility of forming hydrogen bonds with the carboxyl groups of the 1,8-naphthalimide structure, the molecules in the excited state are stabilized and the fluorescence maxima do not follow the trend of dependence on the polarity of the solvents.

In the nonpolar solvent toluene, in addition to the fluorescence maximum at 421 nm, a second maximum appears at λ_F_ = 558 nm, shifted bathochromically relative to the first (Figure 3). In this solvent, the 1,8-naphthalimide fluorophores are situated enough to each other to result in an excimer fluorescence emission due to a specific rearrangement of the 1,8-naphthalimide fragments in the dendrimer molecule. Probably the dipole–dipole interactions between the dendrimer molecules and the polar solvents stabilize the excited state so that the dendrimer emits only monomeric fluorescence. When equimolar amounts of HCl are added to the solution, only the monomer spectra are observed, while the excimer fluorescence disappears. In this case, protonation of the tertiary nitrogen atoms by the receptor fragments results in the reorganization of the fluorophore units and the predominance of monomeric fluorescence at λ_F_ = 418 nm.

The molar extinction coefficient (ε) is at the region 201,600 ÷ 227,000 L mol^−1^ cm^−1^. It is approximately sixteen times higher than that of monomeric 1,8-naphthalimides with the same substituent at the C-4 position [38]. This in turn means that all the primary amino groups of the initial dendrimer have reacted with 1,8-naphthalimide.

### 2.4. Influence of pH on the Fluorescent Intensity

Due to the presence of -OCH_2_CH_2_N(CH_3_)_2_ groups, the fluorescence intensity is also dependent on the polarity of the medium, and high values for the fluorescence quantum yield (Φ_F)_ were obtained in nonpolar solvents, which is explained by photoinduced electron transfer processes that are favored in the polar environment (Figure 4). The lowest Φ_F_ was found in acetonitrile solution (Φ_F_ =0.011) and its value increased more than 71 times in nonpolar solvent tetrahydrofuran (Φ_F_ =0.786). On the other hand, the effect of the formation of hydrogen bonds between the hydroxyl groups of proton donor solvents such as alcohols and the carboxyl groups of the 1,8-naphthalimide structure is superimposed, resulting in a decrease in fluorescence intensity.

The results show that the nature of the solvents affects their spectral characteristics through their polarity and the possibility of specific fluorophore–solvent interactions leading to a change in the polarity of the chromophore system.

The influence of the concentration of protons on the fluorescence intensity of the dendrimer in ethanol–water solutions in a ratio of 1:4, during titration with hydrochloric acid and sodium hydroxide, was investigated. In an alkaline medium, the emitted fluorescence is weak due to the implementation of the photoinduced electron transfer process, as seen in Figure 5. With the increase in the concentration of protons in the medium, an amplification of the fluorescence intensity is achieved until reaching a certain value, which is due to the protonation of the tertiary nitrogen atom of the substituent in the fourth position of the 1,8-naphthalimide nucleus in an acidic environment. As a consequence, the energy of its HOMO orbital is lowered. When irradiated with light, the molecule goes into an excited state, but electron transfer from the nitrogen atom to the 1,8-naphthalimide nucleus does not take place, and the molecule emits fluorescence during its transition from the excited state to the ground state. Upon titration of the dendrimer solution with acid, the fluorescence intensity sharply increases in the range of pH = 10.0–5.5, after which the curve forms a plateau and with a further increase in the proton concentration, the fluorescence intensity does not change. The difference in fluorescence emission at the transition from alkaline to acidic medium is about 50-fold, indicating very good fluorescence switching by varying the pH values of the medium.

The value of the acidity constant (pKa) was calculated by using Equation (1).
pH = pKa + log(I_Fmax_ − I_F_)/(I_F_ − I_Fmin_)(1)

The value for photolith constant pKa = 7.19 is close to the physiological pH in living organisms.

### 2.5. Metal Ions Detection

To clarify the sensing activity of the dendrimer, its fluorescence intensity was investigated in the presence of Pb^2+^, Zn^2+^, Mg^2+^, Sn^2+^, Ba^2+^, Ni^2+^, Sn^2+^, Mn^2+^, Co^2+^, Fe^3+^, and Al^3+^, which are some of the most common environmental pollutants, or their concentration needs to be monitored as biologically important metal ions in living organisms and plants. DMF was used as a solvent, since it is a good solvent for both the dendrimer as the starting ligand and the metal salts and their complexes. Moreover, in the DMF solution, the dendrimer has a low quantum yield fluorescence (FF). When investigating the influence of metal cations on the photophysical characteristics of the dendrimer, it was found that the fluorescence intensity increases in the presence of metal ions, and this effect depends on the nature of the analyzed metal ions (Table 2). The fluorescence intensity enhancement factor (FE) was determined by the ratio of the maximum fluorescence intensity in the presence of metal cations (I) and the initial fluorescence intensity in the absence of cations (I_0_) in the solution (FE = I/I_0_).

From the data in Table 2, it can be seen that the change in the position of the absorption (Δλ_A_ = 6 nm) and fluorescence (Δλ_F_ = 8 nm) maxima is weakly dependent on the type of metal cation. Significantly more coherent is the difference in the fluorescence quantum yields. The lowest values were obtained in the presence of Mg^2+^, Ba^2+^, and Mn^2+^ ions, and the highest in ions Fe^3+^, Al^3+^, Pb^2+^, Zn^2+^, Sn^2+^, Ni^2+^, Cu^2+^, and Co^2+^ ions. The results indicate that the dendrimer ligand forms stable coordination bonds with them and can be used for their detection in non-aqueous media.

Typical changes in dendrimer fluorescence intensity induced by increasing Fe^3+^ ion concentration as an example are presented in Figure 6A. As the concentration of Fe^3+^ ions increases to 1.8 × 10^−5^ mol L^−1^, the fluorescence intensity grows. Its amplification confirms the formation of coordination bonds of iron ions with the receptor fragment at C-4 of the naphthalene nucleus. The obtained linear dependence with R = 0.99983 of the fluorescence intensity of the dendrimer on the concentration of Fe^3+^ ions in the range 0 ÷ 1.8 × 10^−5^ mol L^−1^ is shown in Figure 6B. Using a linear regression method and the equations LOD = 3Sa/b and LOQ = 10Sa/b, where Sa is the standard deviation of the response and b is the slope of the calibration curve [39], determined the limit of quantification (LOQ) and limit of detection (LOD). The obtained values for LOD = 0.997 × 10^−7^ mol L^−1^ and LOQ = 3.17 × 10^−7^ mol L^−1^ are in the sub-ppm concentration range for the detection of Fe^3+^.

Figure 2 shows the supramolecular system “Fluorophore-Spaser-Receptor” for the detection of metal ions as “guest” and the modified dendrimer as “host”. The interaction of the receptor (*N,N*-dimethylamino group) and the fluorophore (1,8-naphthalimide units) causes photoinduced electron transfer; as a result, the fluorescence emission of the dendrimer is quenched and the system is in the ON state. The coordination of the metal ions with the free electron pairs at the receptor nitrogen atoms reduces their donor potential, and this, in turn, causes a reduction or complete shutdown of the electron transfer, resulting in the formation of a fluorescent emitting complex with an “OFF state” of the PET in the system.

### 2.6. Effect of Light Irradiation on Bacterial Growth

The effect of light on the antimicrobial activity of the investigated compound against the model strains was tested in MPB. As can be seen in Figure 7, the compound inhibited Gram-positive *B. cereus* slightly more effectively than Gram-negative *P. aeruginosa* as compared to negative control. The antimicrobial effect increased after light irradiation. In *B. cereus*, at a concentration of 40 µg/mL, about 60% inhibition was established in the dark and about 1.7-fold higher inhibition (76%) in the illuminated sample. *P. aeruginosa* showed higher resistance to the compound than *B. cereus*: at a concentration of 40 µg/mL, the compound inhibited the growth of *P. aeruginosa* by 47% in the dark and by 62% after light illumination.

The increased antimicrobial activity of the studied photoactive dendrimer after light illumination can be explained by the ability to bind to bacterial membranes and generate highly reactive singlet oxygen (^1^O_2_) particles [40,41,42]. Generated reactive oxygen species attack the external layer of the bacterial membrane by multi-target action. Thus, oxidative stress causes irreparable damage to the cellular bacterial components leading to their inactivation [43,44].

## 3. Materials and Methods

The synthesis and characterization of 4-nitro-1,8naphthalimide modified dendrimer have been described [45]. 2-*N,N-*dimethylaminoetanol was used as obtained from Sigma-Aldrich, Munich, Germany. All organic solvents were of spectroscopic grade and were used as obtained from Sigma-Aldrich, Germany without purification. Absorption and fluorescent spectra were taken on Varian Cary 5000 UV-Vis-NIR (Varian Inc., Cary, NC, USA) and Cary Eclipse spectrophotometer (Hellma, Müllheim in Markgräflerland, Germany). ^1^H NMR (600.13 MHz) and ^13^C NMR (150.92 MHz) spectra were acquired on an BRUKER AVANCE AV600 II+NMR spectrometer (Bruker Biospin GmbH. Rheinstetten, Germany) in a dimethyl fulfoxide (DMSO-d_6_) solution at ambient temperature. IR spectroscopic analysis was performed using an IRAffinity-1 spectrophotometer (Shimadzu Co., Kyoto, Japan). To keep a constant total volume during the investigation of the effect of pH on the absorption and emission spectra, small amounts of HCl and NaOH were added. A stock solution of dendrimer was prepared at a concentration of 10^−3^M in DMF and 30 μL was taken from this and placed in 3 mL, yielding solutions at a concentration of 10^−6^ M. Quinine bisulfate/1N H_2_SO_4_ (Φ_f_ = 0.546) was used for the calculation of fluorescence quantum yield by Equation (2).
(2)$$\Phi_{F}=\Phi_{st}\frac{S_{u}}{S_{st}}\frac{A_{st}}{A_{u}}\frac{{n_{Du}}^{2}}{{n_{Dst}}^{2}}$$
where Φ*_F_* is the quantum yield of the dendrimer; Φ*_st_* = 0.546 is the quantum yield of the standard; A*_st_* and A*_u_* represent the absorbance of the standard and dendrimer at the excited wavelength, respectively; while S*_st_* and S*_u_* are the integrated emission band areas of the standard and sample dendrimer, respectively, and not and n_u_ is the solvent refractive index of the standard and sample; subscripts *u* and *s* refer to the unknown (dendrimer) and standard, respectively.

### 3.1. Synthesis of 4-N,N-Dimethylaminoethyloxy-1,8-naphthalimide-Labelled PAMAM Dendrimer

A mixture of 4-Nitro-1,8-naphthalimide-modifed PAMAM dendrimer (0.685 g, 0.10 mM) (0.2 mL, 0,27 mM) 2-*N,N-*dimethylaminoethanol and 0.07 g 0.50 mM K_2_CO_3_ was dissolved in 50 mL *N,N*-dimethylformamide. The mixture was sired at 50 °C for 8 h. After that, the liquor was added to 200 mL ice water and the precipitate was filtered off, washed with water, and dried in the air, Yield: 84%.

FT-IR: cm^−1^: 3067, 2977, 2813, 1684, 1651, 1580, 1526, 1383, 1347, 1268, 1082, 777, 761. ^1^H-NMR (DMSO-d_6_): δ 1.86 (96H, N*CH_3_*), 2.14 (24H, OCNHCH_2_*CH_2_*N), 2.26 (56H, OC*CH_2_*CH_2_N), 2.44 (32H, Ar-ArOCH_2_*CH_2_*N(CH_3_)_2_), 2.98 (56H, N*CH_2_*CH_2_CO), 3.06 (4H, N*CH*_2_*CH_2_*N), 3.12 (24H, CONH*CH_2_*CH_2_N), 3.3029 (32H, (OC)_2_NCH_2_*CH_2_*NHCO), 3.40 (32H, Ar-O*CH_2_*CH_2_N(CH_3_)_2_), 4.12 (32H, (OC)_2_N*CH_2_*CH_2_NHCO), 7.24 (16H, Ar-H), 7.58 (16H, Ar-H), 7.76 (12H, CO*NH*CH_2_CH_2_N), 7.84 (16H, (OC)_2_NCH_2_CH_2_*NH*CO), 8.08 (16H, Ar-H), 8.32 (16H, Ar-H), 8.44 (16H, Ar-H); ^13^C-NMR (DMSO-d_6_): δ 31.3 (NCH_2_*CH_2_*CO), 35.8 (CONH*CH_2_*CH_2_N), 38.0 ((OC)_2_N*CH_2_*CH_2_NHCO), 40.7 (Ar-O*CH_2_*CH_2_N(CH_3_)_2_), 43.9 (N*CH_3_*), 50.4 (N*CH_2_*CH_2_CO), 52.8 (CONHCH_2_*CH_2_*N), 58.9 (Ar-OCH_2_*CH_2_*N(CH_3_)_2_), 103.5- 134.0 (Ar-C), 162.9 and 163.8 (*CO*NH*CO*), 172.3 and 171.9 (*CO*NH).

Elemental analysis: C_398_H_496_O_60_N_58_ (7052.54.): Calcd. C 67.78 H 7.09 N 11.53; Found C 67.53 H 7.18 N 11.62.

### 3.2. In Vitro Biological Tests of the Microorganisms

The antimicrobial activity of dendrimer was tested against two representative strains used as models: Gram-positive *Bacillus cereus* ATCC 11778 and Gram-negative *Pseudomonas aeruginosa* 1390 (Collection of the Institute of Microbiology, Bulgarian Academy of Sciences). Bacterial cultures were maintained at 4 °C on meat-peptone agar slants and transferred monthly. The ability of the dendrimer to inhibit the growth of the model strains was tested in meat-peptone broth (MPB) in dark and under light illumination. The compound was dissolved in DMSO at a started concentration of 1.0 mg/mL and further diluted in test tubes with MPB to final concentrations 40, 25, and 10 µg/mL. After inoculation with each standardized cell suspension (0.5 McFarland standard), the tubes were incubated for 18 h at an appropriate temperature in the presence of light and dark. Positive controls (compound and MPB, without inoculum) and negative controls (MPB and inoculum, without compound) were also prepared. Microbial growth was assessed by the turbidity of the medium at 600 nm (OD_600_). The experiments were conducted in triplicate and the averages were taken (standard deviations less than 5%). A 150 W xenon lamp with a spectral window of the Newport solar simulator bulb (185–1100 nm) at 25 cm sample distance has been used for irradiation of bacterial strains.

## 4. Conclusions

A new second-generation PAMAM dendrimer modified with 4-*N,N*-dimethylaminoethoxy-1,8-naphthalimides units emitting blue fluorescence was synthesized and characterized. The influence of the polarity of the solvents on its photophysical characteristics has been investigated and a hypsochromic shift of the absorption and fluorescence maxima was found upon the transition from non-polar to polar solvents. The fluorescence quantum yield also shows a significant dependence on the type and polarity of the solvents. The change in fluorescence intensity of the dendrimer as a function of pH in a water–alcohol solution at a ratio of (4:1 *v*/*v*) was studied. It was found that in the interval pH = 10.0–5.5, the fluorescence intensity increased more than 50 times, indicating excellent pH fluorescence switching and control of photoinduced electron transfer, which is due to the protonation of the tertiary nitrogen atom of the receptor fragment of 4-*N,N*-dimethylaminoethoxy substituent. The sensor ability of dendrimer to detect different metal cations (Pb^2+^, Zn^2+^, Mg^2+^, Sn^2+^, Ba^2+^, Ni^2+^, Cu^2+^, Mn^2+^, Co^2+^, Fe^3+^, and Al^3+^) has also been investigated. In the presence of some of these metal ions, the intensity of the emitted fluorescence is multiplied, and this effect is best expressed with Fe^3+^ and Al^3+^ ions. According to the order of their activity, the investigated metal ions can be arranged as follows: Fe^3+^> Al^3+^ > Zn^2+^ ~ Pb^2+^ > Cu^2+^> Co^2+^> Sn^2+^ >Sn^2+^ > Ni^2+^ > Mg^2+^ ~ Ba^2+^ ~ Mn^2+^. The antimicrobial activity of the dendrimer was tested in vitro against Gram-positive *B. cereus* and Gram-negative *P. aeruginosa* in a meat-peptone agar medium in the dark and after light irradiation. It was established that after light irradiation, the dendrimer’s antibacterial activity increases due to the generation of reactive oxygen species that attack the cell membranes of bacteria. In this case, *B. cereus* appears to be slightly more sensitive than *P. aeruginosa*.

## Data Availability

The original contributions presented in the study are included in the article, further inquiries can be directed to the corresponding authors.

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
