# Peer review of "Sensing and Microbiological Activity of a New Blue Fluorescence Polyamidoamine Dendrimer Modified with 1,8-Naphthalimide Units"

_molecules, 2024, doi:10.3390/molecules29091960_

Round 1

Reviewer 1 Report

Comments and Suggestions for Authors

The authors synthesized a novel second-generation blue fluorescent polyamidoamine dendrimer with blue fluorescence. The photophysical characteristics were investigated in organic solvents of different polarity and pH. The ability of the dendrimer to detect metal ions was also investigated. The antimicrobial activity of the new compound has been tested in vitro against Gram-positive B. cereus and Gram- negative P. aeruginosa. The manuscript needs some improvement before publishing.

1. The synthesis process was not clear, and the mass spectrogram is needed.

2. In their presence of metal ions, the fluorescent intensity was amplified more than 66 times. For each type of ion, both the detection limit and the increase in fluorescence intensity need to be provided.

3. It is necessary to provide the absolute fluorescence quantum yield of the material in each solvent, rather than relative.

4. The energy levels of singlet and triplet states of luminescent materials need to be characterized to understand the process of photodynamic antibacterial action.

5. Electrochemical oxidation and reduction are suggested for testing, along with an analysis of the material's frontier orbital energy levels.

6. The dynamic study of changes in probe fluorescence intensity before and after can be analyzed through transient fluorescence spectroscopy.

Comments on the Quality of English Language

Moderate editing of English language required.

Author Response

  1. The synthesis process was not clear, and the mass spectrogram is needed.

What exactly is not clear about dendrimer synthesis? This is a routine procedure for nucleophilic substitution of a nitro group in 1,8-naphthalimide in DMF medium. For the dendrimer characterization, we have used H NMR and C NMR, IR and elemental analysis, which we believe is sufficient. Mass spectrometry was not used.

  1. In their presence of metal ions, the fluorescent intensity was amplified more than 66 times. For each type of ion, both the detection limit and the increase in fluorescence intensity need to be provided.

The indicated fluorescence enhancement (FE) of 66 times refers to the presence of Fe ions. The fluorescence enhancement in the presence of the other metal ions is given in Table 2, where the fluorescence quantum yields for each metal ion are also indicated. The detection limit of iron ions is provided as an illustrative example.

  1. It is necessary to provide the absolute fluorescence quantum yield of the material in each solvent, rather than relative.

It is a common and routine procedure to characterize fluorescent compounds by relative quantum yield against known standards,  by equation 1.

  1. The energy levels of singlet and triplet states of luminescent materials need to be characterized to understand the process of photodynamic antibacterial action.

 The synergism of light and the dendrimer's biological activity has been used to study its antibacterial activity. For this reason, it is not necessary to establish these parameters.

  1. Electrochemical oxidation and reduction are suggested for testing, along with an analysis of the material's frontier orbital energy levels.

Electrochemical oxidation and reduction were not the subject of this study.

  1. The dynamic study of changes in probe fluorescence intensity before and after can be analyzed through transient fluorescence spectroscopy.

It is not clear „before and after“…..?

Reviewer 2 Report

Comments and Suggestions for Authors

Dear Authors,

I have thoroughly assessed your manuscript titled " Sensing and microbiological activity of a new blue fluorescence polyamidoamine dendrimer modified with 1,8-naphthalimide units" and commend you for your dedication to advancing scientific knowledge. However, I have identified several areas that could benefit from improvement:

1.     High-Resolution Mass Spectrometry (HRMS): Please include HRMS data to provide a comprehensive characterization of the dendrimer.

2.     High-Performance Liquid Chromatography (HPLC) Purity: It would be beneficial to include HPLC purity data for the dendrimer to ensure the accuracy of your findings.

3.     Time-Dependent Antimicrobial Activity Experiment: Consider conducting a time-dependent antimicrobial activity experiment with the following steps:

(i) Irradiation of microbes with various light doses (15 min, 30 min, 1h, 2 h, 5 h, 10h),  while keeping another batch of microbes in the dark for the same duration as a reference.

(ii) Growth of microbes in the dark followed by the calculation of microbial growth.

Addressing these points will significantly enhance the quality and impact of your manuscript. Thank you for your dedication to scientific research and knowledge advancement.

Author Response

We thank you for reviewing our manuscript and making constructive suggestions. Our responses, albeit negative, are given below.

  1. High-Resolution Mass Spectrometry (HRMS): Please include HRMS data to provide a comprehensive characterization of the dendrimer.

Unfortunately, due to the high molecular mass and not very good solubility of the dendrimer, we could not use HRMS for its characterization. We have used the classical H NMR and C NMR and IR spectral methods.

  1. High-Performance Liquid Chromatography (HPLC) Purity: It would be beneficial to include HPLC purity data for the dendrimer to ensure the accuracy of your findings.

For the same reason, HPLC was not used. Elemental organic analysis was used for the purity of the dendrimer.

  1. Time-Dependent Antimicrobial Activity Experiment: Consider conducting a time-dependent antimicrobial activity experiment with the following steps:

(i) Irradiation of microbes with various light doses (15 min, 30 min, 1h, 2 h, 5 h, 10h),  while keeping another batch of microbes in the dark for the same duration as a reference.

(ii) Growth of microbes in the dark followed by the calculation of microbial growth.

This recommendation is very interesting and we will keep it in mind in our future research. Due to the specificity of the journal, we have focused more on the synthesis, photophysical properties and sensing activity of the dendrimer and have only shown preliminary studies of its antimicrobial activity. In-depth biophysical studies are pending to clarify the mechanism of interaction of the dendrimer with the bacterial membrane.

Round 2

Reviewer 1 Report

Comments and Suggestions for Authors

The article can be published in this form. 

Comments on the Quality of English Language

The English presentation needs professionals to improve.

Author Response

Thanks to the reviewer for his comments.

Reviewer 2 Report

Comments and Suggestions for Authors

After thoroughly evaluating the manuscript titled "Sensing and microbiological activity of a new blue fluorescence polyamidoamine dendrimer modified with 1,8-naphthalimide units," it is evident that the authors have successfully synthesized a blue fluorescent polyamidoamine dendrimer and investigated its photophysical properties, metal ion detection ability, and antimicrobial activity.

However, to improve the characterization of the synthesized compound, I recommend that the authors provide the NMR graph and elemental analysis report as mentioned in their response to the reviewer's comment. Including this additional data would significantly enhance the clarity and completeness of the manuscript. Thank you.

Author Response

After thoroughly evaluating the manuscript titled "Sensing and microbiological activity of a new blue fluorescence polyamidoamine dendrimer modified with 1,8-naphthalimide units," it is evident that the authors have successfully synthesized a blue fluorescent polyamidoamine dendrimer and investigated its photophysical properties, metal ion detection ability, and antimicrobial activity.

However, to improve the characterization of the synthesized compound, I recommend that the authors provide the NMR graph and elemental analysis report as mentioned in their response to the reviewer's comment. Including this additional data would significantly enhance the clarity and completeness of the manuscript. Thank you.

Thanks to the reviewer for his comments. Elemental analysis data were obtained directly from the elemental analyzer and are given in the experimental section following the description of the dendrimer synthesis. In the experimental part, data from the NMR spectra describing the structure of the dendrimer are also given.